# Biomarkers of Post-COVID Depression

**DOI:** 10.3390/jcm10184142

**Published:** 2021-09-14

**Authors:** Piotr Lorkiewicz, Napoleon Waszkiewicz

**Affiliations:** Department of Psychiatry, Medical University of Bialystok, Plac Brodowicza 1, 16-070 Choroszcz, Poland; napwas@wp.pl

**Keywords:** COVID-19, depression, biomarkers, post-COVID, kynurenine, cytokines

## Abstract

The COVID-19 pandemic is spreading around the world and 187 million people have already been affected. One of its after-effects is post-COVID depression, which, according to the latest data, affects up to 40% of people who have had SARS-CoV-2 infection. A very important issue for the mental health of the general population is to look for the causes of this complication and its biomarkers. This will help in faster diagnosis and effective treatment of the affected patients. In our work, we focused on the search for major depressive disorder (MDD) biomarkers, which are also present in COVID-19 patients and may influence the development of post-COVID depression. For this purpose, we searched PubMed, Scopus and Google Scholar scientific literature databases using keywords such as ‘COVID-19’, ‘SARS-CoV-2’, ‘depression’, ‘post-COVID’, ‘biomarkers’ and others. Among the biomarkers found, the most important that were frequently described are increased levels of interleukin 6 (IL-6), soluble interleukin 6 receptor (sIL-6R), interleukin 1 β (IL-1β), tumor necrosis factor α (TNF-α), interferon gamma (IFN-γ), interleukin 10 (IL-10), interleukin 2 (IL-2), soluble interleukin 2 receptor (sIL-2R), C-reactive protein (CRP), Monocyte Chemoattractant Protein-1 (MCP-1), serum amyloid a (SAA1) and metabolites of the kynurenine pathway, as well as decreased brain derived neurotrophic factor (BDNF) and tryptophan (TRP). The biomarkers identified by us indicate the etiopathogenesis of post-COVID depression analogous to the leading inflammatory hypothesis of MDD.

## 1. Introduction

The coronavirus disease (COVID-19) caused by the SARS-CoV-2 virus has been spreading worldwide for the last 1.5 years. According to WHO data, over 187 million cases have been diagnosed globally, including over 4 million fatalities [1]. 

The most common symptoms of coronavirus disease are fever, cough, shortness of breath, muscle pain, headache, diarrhea, rhinorrhea, loss of smell and taste [2,3,4]. In addition, there are more and more reports of mental health problems in people who have survived SARS-CoV-2 infection. The most frequently described mental disorders are major depressive disorder (MDD), post-traumatic stress disorder (PTSD), anxiety disorders, obsessive-compulsive disorders (OCD) and insomnia [5,6,7]. These disorders occur mainly in the acute phase of infection and shortly after it [7,8,9]. While the symptoms of PTSD, anxiety disorders and insomnia gradually disappear, it has been shown that symptoms of MDD persist even in the third month of follow-up [7]. More than 75% of COVID-19 patients have cognitive difficulties with episodic memory, attention and concentration, which is also common state in MDD (called pseudo-dementia), and these might occur even after mild infection [10]. Even two to three months after disease onset, patients also had deficits in executive functions and visuospatial processing [11]. The large number of people infected with SARS-CoV-2 and the prevalence of MDD among those who have experienced COVID-19—according to some data, ranging from 28% to 45% [6,12]—may contribute to the emergence of a serious global problem and significantly increase the pool of people suffering from major depressive disorder.

The COVID-19 crisis is about more than general health problems. Prolonged disease symptoms, employees’ inability to work, the closing of businesses, premature deaths and the cost of COVID-19 prevention and treatment make most of the countries affected by the pandemic feel the economic crisis as well. In the United States alone the total cost of the pandemic is estimated at approximately 90% of the annual gross domestic product, which is more than $16 trillion [13]. Therefore, the search for different ways to limit the effects of the pandemic is of great importance for the economic sector as well. 

Depression, also called major depressive disorder (MDD) is the most common psychiatric disease in the world and one of the most common causes of disability measured in years lived with the disease (YLDs) [14]. According to WHO, over 264 million people currently struggle with MDD worldwide [15]. So far, the diagnosis of MDD is based mainly on clinical symptoms and scales [16,17], although for several years the field of psychiatry has been searching for useful biomarkers of MDD, which could allow to detect the disease, implement treatment faster and monitor its efficiency in a more objective way [18,19].

In recent years, many new theories about the pathophysiology of major depressive disorder have emerged. They include disorders in systems such as the immune [20,21], endocrine [22,23] or digestive [24,25], as well as changes in the metabolome [26,27], neurotrophic factors [28,29] or oxidative stress [30,31]. Although the research on the above mentioned theories has led to the proposition of many promising biomarkers of MDD [32,33,34], further research is still required as the results of many studies often differ [35,36,37].

Nowadays, facing the danger posed by the SARS-CoV-2 coronavirus pandemic, we should do our best to counteract its consequences [38]. These consequences also include post-COVID depression [7,39].

The aim of the study is to find a potential link between biomarkers of MDD and markers of disturbed homeostasis of the organism during or after COVID-19. Common links of those two states can be of use as prognostic biomarkers of post-COVID depression and indicate the cause of MDD development in those patients. The acquired knowledge could be used in the future to determine which COVID-19 (+) patients are at risk of developing MDD or making a more confident diagnosis in those who are already affected. This will allow to identify the group of patients in whom mental health should be paid special attention in the phase of recovery from illness and during regaining social functionality from before SARS-CoV-2 infection. Doing so would enable the implementation of effective treatment faster which will reduce recovery time for patients, reduce treatment costs and economic burden of COVID-19 and contribute to some extent to reducing the global mental health problem.

## 2. Materials and Methods

A literature search was conducted in PubMed, Scopus and Google Scholar databases. We included clinical studies, reviews, meta-analyzes and case studies regarding depressive symptoms in COVID-19 (+) patients. The search strategy consisted of the following keywords: ‘depression’, ‘biomarker’, ‘COVID-19’, ‘SARS-CoV-2’, ‘post-COVID’, ‘long-COVID’, ‘metabolomic’, ‘inflammatory’, ‘immunological’, ‘endocrinological’ ‘oxidative stress’, ‘HPA axis’, ‘neurotrophic’, ‘biosignature’ as well as combinations of these terms. We then excluded all articles in which MDD was described in the context of the social consequences of a pandemic such as isolation, stress, fear of disease or economic problems, in order to be able to analyze only the direct impact of SARS-CoV-2 on the body and the development of post-COVID depression. Relevant studies were then included with the intention of covering the widest possible spectrum of different markers for post-COVID depression. We conducted additional manual searches of the references of the related articles in order to gather information about the relevant supporting literature.

## 3. Results

Taking into account the biological systems involved in the development of MDD and various theories of its pathophysiology [37,40], in this study we divided biomarkers into inflammatory, kynurenine pathway and growth factors.

### 3.1. Inflammatory Factors

The role of inflammation and inflammatory factors in the development of MDD is well understood [41,42]. It has been proven that patients suffering from inflammatory diseases, such as multiple sclerosis or systemic lupus, who have also been treated with cytokines have a greater chance of developing MDD [43,44,45,46]. COVID-19 is a disease that can cause systemic inflammation and a cytokine storm [47], and is therefore analogous to inflammatory diseases, and may contribute to the development of major depressive disorder (MDD).

Inflammatory cytokines are molecules that mediate the immune response upon activation of the peripheral immune system. Cytokines such as tumor necrosis factor alpha (TNFα), interferon-γ (IFN-γ), interleukin 1 (IL-1) play a role primarily in enhancing the cellular response, while cytokines such as interleukin 6 (IL-6), interleukin 10 (IL-10), interleukin 13 (IL-13) are more associated with the humoral response [41,48]. Cytokines can be produced in the brain by astrocytes and microglia [49,50] or reach it from the periphery due to several mechanisms such as passage through leaky blood-brain barrier (BBB) regions, including the choroid plexus and periventricular organs, active transport through cytokine transport molecules on the endothelium of the brain, transmission of cytokine signals during an infection in the abdominal cavity via afferent nerve fibers such as the vagus nerve, the passage of activated monocytes into the brain from the periphery, or by signals of second-messengers from the BBB endothelial lining which results in overproduction of cytokines [51,52,53].

Moreover, the activated inflammatory process and its mediators, such as TNFα, cause changes in the blood-brain barrier by affecting the endothelial cells forming the barrier, which in turn leads to its increased permeability [54]. This makes it easier for other cytokines and inflammatory factors to penetrate into the brain.

Cytokines exert a number of actions in the brain that are related to the development of MDD, including: activation of the hypothalamic-pituitary-adrenal axis and induction of the resistance to glucocorticoids [55,56]. They also cause disturbances in the neurotransmitter system [57,58], affect neuroplasticity [59] and hippocampal neurogenesis, as well as disturb the neurotrophin signaling cascade [41].

Due to the nature of the disease, there are changes in the cytokine system that are most often described in COVID-19 [47,60]. Many of them coincide with changes in MDD and may be the cause of post-COVID depression. Below we present the most frequently described major depressive disorder biomarkers that coincide with those found in patients in the acute phase of COVID-19 or during follow-up (Figure 1).

#### 3.1.1. IL-6/sIL-6R

Interleukin 6 (IL-6) is a pro-inflammatory pleiotropic cytokine secreted mainly by monocytes and macrophages under the influence of interleukin 1β and TNF-α, but also by astrocytes and microglia. It belongs to the family of proteins that use gp130 as a signal transmitter [61]. It works by inducing the differentiation of activated B cells towards antibody-secreting cells, stimulating the synthesis of acute phase proteins such as C-reactive protein, serum amyloid A, fibrinogen, α1-antitrypsin and haptoglobin in the liver, and promoting the differentiation of naive CD4 + T cells [62]. It also plays a role in the body as a mediator of a warning signal about tissue damage or other sudden events. Its level increases in the event of infection, inflammation or trauma [61]. IL-6 also affects the functionality of neurons, may disrupt hippocampal neurogenesis and intensifies neuroinflammation [63]. IL-6 has two ways of interaction, the classical one, i.e., anti-inflammatory, and the trans-signaling or pro-inflammatory one [64]. IL-6 exerts its anti-inflammatory function after binding to the membrane-bound IL-6 receptor (mIL-6R) present on the cell membranes of several types of cells, e.g., certain subtypes of T cells, hepatocytes, neutrophils, megakaryocytes or monocytes. The pro-inflammatory pathway of IL-6 becomes activated upon binding to the soluble IL-6 receptor (sIL-6R) and formation of the IL-6/sIL-6R complex, which affects all cells expressing gp130 [64].

It is the cytokine most often described in MDD, and the studies describing its involvement in MDD are mostly consistent with each other [32,34,41,65,66]. According to some studies, high levels of IL-6 correlate with the severity of MDD symptoms in patients who do not respond to treatment [67]. It has been observed that in patients with MDD, increased levels of IL-6 correlated with attention deficit disorder, and one study showed that its increase was prior to the incidence of cognitive impairment in patients with MDD [66].

High levels of IL-6 is widely described in relation to COVID-19, and its level corresponds to the severity of the disease [7,68,69,70,71]. In one study, high levels of sIL-6R were also detected in few COVID-19 (+) patients, but the results are not consistent and more research is needed [71]. Higher levels of IL-6 in tandem with higher levels of sIL-6R lead to increased trans-signaling [72]. Thus, if the severity of COVID-19 correlates with the amount of IL-6, patients with severe course may be at greater risk of developing post-COVID depression.

#### 3.1.2. IL-10

Interleukin 10 (IL-10) is an anti-inflammatory cytokine produced by Th2, Treg cells and M2 macrophages [73]. In the central nervous system (CNS) it is produced, inter alia, by astroglia and microglia [74]. In the latter, the secretion of IL-10 is augmented by neurotransmitters and damage-associated molecules—glutamate and adenosine respectively [75]. Overall IL-10 production increases with increased levels of IL-6 and TNF-α [76,77]. It exerts its function by binding with IL-10 receptor (IL-10R) which consist of two subunits—IL-10R_1_ and IL-10R_2_. The latter is expressed in most cell types, but IL-10R_1_ is mostly restricted to cells of hematopoietic lineage. Due to the myeloid origins, microglia express both subunits of IL-10R. Unexpectedly, resting astrocytes also express IL-10R_1_ [75]. IL-10 limits neuroinflammation, promotes production of immunosuppressive transforming growth factor β (TGF-β) by astrocytes and reduces astrogliosis in response to the pathogenic factors [75].

Elevated levels of IL-10 are often reported in people with MDD [78,79,80]. Some researchers also observed that MDD severity is related to increased IL-10 [81,82]. Two meta-analyzes have shown that IL-10 level decreases with effective antidepressant treatment [78,83].

To date, many studies have described an increased level of IL-10 in COVID-19 patients [84,85,86]. In severe cases, the level of IL-10 was higher than in mild cases, and positively correlated with mortality due to COVID-19 [47,85,87,88,89]. Thus, high levels of IL-10 during and after SARS-CoV-2 infection may suggest an increased susceptibility to developing MDD in these patients. It may also be a good indicator for monitoring the treatment of post-COVID depression, due to its decline with antidepressant treatment [78].

#### 3.1.3. TNF-α/sTNFR1, sTNFR2

Tumor necrosis factor α (TNF-α) is a pro-inflammatory cytokine produced by Th1 lymphocytes and M1 macrophages, and by astroglia and microglia in the brain [32]. It is one of the earliest cytokines released following trauma, infection or exposure to lipopolysaccharide (LPS) [90] and a regulator of pro-inflammatory cytokine production. Its high level induces the production of, among others, CRP, IL-6, IL-1β [90,91,92].

It is found to be one of the most promising markers of major depressive disorder. Its elevated level, along with CRP and IL-6, is most consistently described in studies on MDD biomarkers [20,21,32,33,34,35,36,41,42,93]. It is one of the major cytokines involved in neuroinflammation [94] and acts as an inhibitor of hippocampal neurogenesis [95,96], an inducer of apoptosis [97,98] and negatively affects synaptogenesis, synaptic plasticity and the structure of synaptic membranes [99,100]. It increases the permeability of BBB [54] and significantly affects the production of serotonin through its ability to activate indoleamine 2,3-dioxygenase (IDO) [101,102]. 

Its level decreases with effective treatment of MDD [103], and its persistent concentration indicates treatment-resistant depression (TRD) [104,105]. It exerts its effects through the TNF-R1 and TNF-R2 receptors on cell membranes. They can be released into the serum and are elevated in MDD [106,107,108].

TNF-α is elevated in most COVID-19 patients and correlates with the severity of the disease [6,7,68,70,109]. COVID-19 (+) patients requiring intensive care unit (ICU) admission have higher TNF-α levels when compared to the patients who do not require treatment on ICU [85,110]. One study also found that while in cases of sepsis and acute respiratory distress syndrome (ARDS) in COVID-19 (−) patients, TNF-α levels normalized rapidly after the primary immune response [111,112], it is consistently elevated in COVID-19 (+) patients [85]. This may result in an increased chance of developing post-COVID depression due to longer exposure to the pro-inflammatory effects of TNF-α.

In addition, in one of the studies, an increase in the soluble TNF-α receptors—sTNFR1 and TNFR2 in the serum was observed with the increase in the severity of the disease, and the highest levels were recorded in people who eventually died from COVID-19 [87]. However, so far, not many studies have studies described changes in TNF-α receptors levels, so the importance of this biomarker requires further research.

#### 3.1.4. IL-1β

Interleukin 1β (IL-1β) is another pro-inflammatory cytokine secreted by the same cell types as TNF-α and IL-6, i.e., Th1 lymphocytes and M1 macrophages. In the CNS it can be secreted by different types of cells, e.g., astroglia, oligodendrocytes, neurons and microglia, furthermore, they all express IL-1β receptors [32,113]. Its effects on the brain are, as in the case of TNF-α and IL-6, induction of apoptosis [114,115] and negative effects on synaptogenesis and synaptic plasticity [116,117]. Some studies have proved that physiological levels of IL-1β promote long term potentiation (LTP) and acquisition and retention of memory. In the other hand, its pathophysiological levels as seen in inflammation can disturb LTP and cause failure of memory acquisition and its recall [113]. Additionally, it lowers neurogenesis in human hippocampal progenitor cells by activating the kynurenine pathway [118]. IL-1β along with TNF-α and IL-6 can induce the expression of pro-inflammatory genes in astrocytes and increase neuroinflammation and neurodegeneration [119].

Research on the role of IL-1β in MDD is not consistent. Some studies report a correlation and an increase of this cytokine in depression [120,121], although not all researchers have found such a link [78,122]. The level of IL-1β increases with the increase in BMI [123] and the number of depressive episodes [32], therefore it may cause inaccuracies in research and difficulties in obtaining uniform results. There have also been studies on the proportionality of the level of IL-1β to the severity of MDD symptoms [124] and the fact that it is a risk factor for TRD [125,126].

IL-1β has been reported to be elevated in COVID-19 and correlated with disease symptoms [127,128,129]. Severe patients have significantly elevated levels [130], and one study found high levels of IL-1β persistent in COVID-19 patients up to 4 weeks after symptom onset—similar to IL-6 [131]. However, not all studies have shown an IL-1β elevation among COVID-19 patients [132,133], therefore, the importance of this interleukin in post-COVID depression requires more investigation.

#### 3.1.5. IFN-γ

Interferons are a superfamily of endogenous pleiotropic cytokines that play a large role in the maintenance of homeostasis and defense against infection. Interferon gamma (IFN-γ) is a pro-inflammatory cytokine belonging to the type II interferon family [134]. It is secreted mainly by natural killer cells (NK) and CD4 + T cells and macrophages [135].

Disturbances in IFN-γ levels have been documented among patients suffering from MDD [136,137]. Its central or peripheral administration causes symptoms of sickness behavior such as anhedonia, memory and social interaction disorders—the same as seen in MDD [136]. IFN-γ has been shown to activate microglia, which contributes to the development of depression [138]. Moreover, IFN-γ largely activates IDO and contributes to the transition of tryptophan to the kynurenine pathway metabolites which are involved in the pathogenesis of MDD [138,139,140,141].

It has been shown that up to 40% of patients treated with interferon for hepatitis C develop symptoms of depression [139,142].

However, reports on the role of IFN-γ are inconsistent. Some studies demonstrated its increase among people with MDD [140,141,143], and some showed no increase or even a decrease [44,134].

For COVID-19, the research also diverges. Many studies have shown an increase in the concentration of IFN-γ [84,90,91,105,134,139,140,144,145,146], however, several studies showed a decrease [47,147,148]. In some studies, the severity of COVID-19 positively correlated with the level of IFN-γ [88,89,131]. Due to divergent research results, there is still need for more comprehensive studies on this biomarker.

#### 3.1.6. CRP

C-reactive protein (CRP) is an acute-phase protein produced by liver cells in response to injury, infection or inflammation. Its production is induced by IL-6, and IL-1 enhances this effect. During inflammatory diseases, its serum concentration increases by a minimum of 25% [149]. Baseline CRP levels may be influenced by factors such as body weight, sex, age, nicotinism, and lipid levels [150].

Its elevated level in patients with MDD is widely described in the literature [151,152,153,154,155,156,157,158,159], and one of the studies noted that its high level preceded development of a de novo MDD, and therefore, it may be its promising prognostic marker [156]. According to some researchers, its significant increase also correlates with the occurrence of TRD [160,161]. Its higher level is more often detected in the case of atypical depression and is associated with suicidal tendency [158,162]. However, not all studies agree with each other, and some negate its relationship with MDD [163,164], which may be related to many factors that affect its concentration and interfere with its examination solely for MDD.

In the context of COVID-19, an increase in CRP concentration is also often described [7,86,165,166,167,168], and its concentration, according to some reports, correlates with the severity of COVID-19 [165,166,167,169,170].

One study reported an association of post-COVID depression with inflammatory biomarkers, where higher CRP was observed in COVID-19 (+) depressed patients than in COVID-19 (+) patients without depression [171]. A separate study reported a decrease in baseline CRP levels in COVID-19 (+) patients in whom the severity of depression symptoms decreased compared to COVID-19 (+) patients in whom CRP and depression symptoms were not significantly reduced [172]. Unlike the previous two, the study by Mazza et al. in the third month of follow-up in patients with post-COVID depression showed no association with CRP [68].

The relationship of CRP to COVID-19 (+) is well documented, but due to the inflammatory nature of the disease and therefore an overall increase in CRP, its utility as a marker of post-COVID depression requires further investigation.

#### 3.1.7. IL-2/sIL-2R

Interleukin 2 (IL-2) is a pro-inflammatory cytokine produced mainly by CD4 + Th cells and to a lesser extent by T CD8+ and NK cells. It is released mainly in response to an antigen [173]. It exerts its action through IL-2 receptors (IL-2R) present mainly on activated T cells. These receptors can also be released into the bloodstream—soluble IL-2R (sIL-2R). The function of IL-2 is primarily to regulate the function of T cells [173,174].

Studies of MDD biomarkers in affected patients have shown an increased level of this cytokine as well [152,175], and its higher concentration is observed in atypical rather than melancholic depression [175]. The influence of IL-2 on the occurrence of depression symptoms is evidenced by the development of depressive symptoms in people and animals to whom it was administered [136,176,177,178]. Moreover, the concentration of sIL-2R is also elevated in people with MDD [78,151,179,180].

In COVID-19 (+) patients, concentrations of both IL-2 and sIL-2R are higher than in COVID-19 (−) patients, especially in severe patients [47,87,110,147,181,182,183]. Two studies also found an association of increased levels of IL-2 [87] and sIL-2R with mortality from COVID-19 [183], and in another, a high sIL-2R/lymphocyte ratio proved to be the best indicator of critical disease differentiation [184].

There are no studies describing a direct relationship of IL-2R/sIL-2R with post-COVID depression, but due to the clearly described increase in their levels in people with MDD and in people with COVID-19, it may be a promising marker.

#### 3.1.8. MCP1/CCL2

Monocyte Chemoattractant Protein-1/Chemokine ligand 2 (MCP-1/CCL2) is a chemokine produced by many types of cells, e.g., endothelium, fibroblasts, macrophages, monocytes, astrocytes and microglia. This chemokine regulates the migration and infiltration of monocytes, memory T and NK cells at the site of inflammation [185]. MCP-1 has been shown to attract peripheral monocytes to the cerebral cortex, hypothalamus and hippocampus regions, i.e., those that contribute to the development of MDD [186]. Two meta-analyzes showed an increased level of MCP-1/CCL2 in patients with MDD compared to the healthy control group [187,188].

Patients with COVID-19 also show an increased level of this chemokine compared to healthy people, and in a large proportion of cases it is more elevated in people with severe disease compared to patients with a mild form [7,87,110,130,131]. One study also found that MCP-1/CCL2 elevation positively correlated with mortality from COVID-19 [87].

The described pro-depressive effect of MCP-1/CCL2 and its increased level, especially in severe cases of COVID-19, make it relevant to further research this chemokine in terms of the development of post-COVID depression.

#### 3.1.9. SAA1

Serum amyloid A (SAA1) is an acute phase protein that is mainly produced in the liver as a result of IL-1β and IL-6 action [189]. It affects many aspects of the inflammation cascade. by binding to various receptors such as toll like receptors 2 and 4 (TLR2 and TLR4) and the receptor for advanced glycation end products (RAGE) and CD36 [190]. SAA1 activates the secretion of cytokines such as TNF-α, IL-6, IL-8, IL-23, IL-18 and IL-10 [79]. TLR2 and TLR4 receptors are present on macrophages, microglia and astrocytes [79,191], and their stimulation by SAA1 causes production of inflammatory cytokines, including IL-6 and TNF-α, which may play a role in the development of neuroinflammation, which contributes to the occurrence of depressive symptoms [192].

Compared to healthy subjects, patients with symptoms of MDD have elevated levels of SAA1 [79,155,193,194]. Patients admitted to hospital for COVID-19 also show elevated levels of this protein compared to healthy controls [89,195,196]. Additionally, its high concentration positively correlated with the severity of the disease and mortality due to COVID-19. One study noted that a decrease in SAA1 within two weeks of disease was associated with the prognosis of clinical improvement in patients, while its persistently high concentration was associated with death [195].

The clear association of SAA1 with MDD and its persistent high concentration in severe COVID-19 (+) patients suggests that a closer look at its relationship with post-COVID depression is needed.

### 3.2. Kynurenine Pathway

Kynurenine is a tryptophan metabolite formed as a result of its transformation under the influence of the enzymes indoleamine 2,3-dioxygenase (IDO) and tryptophan 2,3-dioxygenase (TDO). Increased activity of these enzymes, in particular IDO, causes the conversion of tryptophan to serotonin (5-HT) to be reduced at the cost of its conversion to kynurenine [197]. Other metabolites of the kynurenine pathway include: kynurenic acid (KYNA), 3-hydroxykynurenine (3-HK), 3-hydroxyanthranilic acid (3-HAA) and quinolinic acid (QA) [197,198]. It has been shown that the decrease in tryptophan level and the accumulation of metabolites of the kynurenine pathway lead to anergy of effector T cells and proliferation of Treg and T, B, and natural killer cell apoptosis [199]. KYNA is a competitive antagonist of the glutamate receptor and an inhibitor of the α7 nicotinic acetylcholine receptor and has a neuroprotective effect [101], 3-HK is associated with neuronal apoptosis, and QA, which is an agonist of the N-methyl-d-aspartate receptor (NMDAR), is associated with excitotoxic neurodegeneration [200]. In the brain, IDO is expressed by astrocytes and microglia, but the metabolites of tryptophan metabolism by IDO differ in these two types of cells. The main metabolite in astrocytes is kynurenic acid (KYNA), and in microglia—quinolinic acid (QA) or 3-hydroxykynurenine (3-HK) [201]. 

IDO activity is stimulated by pro-inflammatory cytokines such as TNF-α, interferons and prostaglandins. IFN-γ has the strongest stimulating effect on IDO [84]. For this reason, a condition with elevated inflammatory mediators, systemic inflammation, serious infection or trauma predispose to tryptophan transfer to the kynurenine pathway and increased production of its neurotoxic metabolites.

The lower level of tryptophan in the blood serum of patients with MDD compared to the serum of healthy controls has been well documented [139,202,203,204,205], but not all reports are consistent and there have been studies in which the levels of tryptophan in patients with MDD did not differ from healthy controls or was even higher in patients with MDD [206,207]. At the same time, in the group of depressed patients there is a higher level of kynurenine metabolites such as kynurenine (KYN), quinolinic acid (QA) and 3-hydroxykynurenine (3-HK), as well as a reduced level of neuroprotective kynurenic acid (KYNA) [151,208,209,210]. In a study by Haroon et al. patients with a high TNF-KYN/TRP ratio showed a greater severity of depression and treatment resistance [210]. In one case, depressed patients with a high QA/KYNA ratio also had a greater severity of associative memory impairment [211].

In a metabolomic study of COVID-19 (+) patients, tryptophan metabolism was the major disorder detected [69]. In patients infected with SARS-CoV-2, decreased levels of tryptophan and serotonin as well as increased levels of kynurenine, 3-hydroxykynurenine, kynurenic acid and picolinic acid (also one of the metabolites of the kynurenine pathway) were found [69,145,212,213]. One of the reasons for the decreased level of tryptophan in COVID-19 (+) patients may also be its decreased absorption in the gut. SARS-CoV-2 causes the internalization and downregulation of ACE2, which is highly expressed in the intestines [214], and which is needed for the expression of the neutral amino acid transporter in the intestinal lumen—B0AT1 [215]. Activation of the kynurenine pathway can be indirectly assessed by the kynurenine to tryptophan ratio, which was significantly higher in COVID-19 (+) patients than in healthy controls and correlated positively with disease severity [69,212,213].

In conclusion, disturbances in tryptophan metabolism and activation of the kynurenine pathway are well described in research on the pathophysiology of MDD and are of great importance in the search for its biomarkers. Analogous changes can be seen in patients infected with SARS-CoV-2 (Figure 2), and at present disturbances in tryptophan metabolism are one of the most promising theories on the development of post-COVID depression [216]. Therefore, it is worth considering metabolites of the kynurenine pathway in future studies for biomarkers of depression developing in patients after SARS-CoV-2 infection, especially in those with a severe course of the disease.

### 3.3. Growth Factors 

#### BDNF

Brain derived neurotrophic factor (BDNF) belongs to the family of neuronal growth factors and its physiological role is to support the differentiation, maturation and survival of dopaminergic, cholinergic and serotonergic neurons of the central nervous system [217]. It also exhibits neuroprotective abilities, is involved in neuroplasticity and enhances long-term potentiation [218,219,220]. It is produced by neurons as well as by peripheral cells such as leukocytes and endothelial cells and is able to pass through the BBB [221]. Its decreased level is a common finding in people suffering from MDD, and effective antidepressant therapy restores it to normal levels [221,222,223].

Previous studies have proved that ACE2 is associated with a reduction in BDNF levels [224]. It is widely believed that SARS-CoV-2, by using ACE2 to enter cells, causes its downregulation [225]. This mechanism may cause a secondary reduction in BDNF levels. The confirmation of this theory may be reflected in one of the studies performed, in which patients suffering from COVID-19 were tested for serum BDNF levels. The researchers demonstrated that patients with moderate and severe disease have lower BDNF levels than those with mild disease, and during patients’ recovery, their levels returned to normal [165].

The association of growth factors, especially BDNF, with MDD is often indicated in the literature (Figure 3), and the likely mechanism by which SARS-CoV-2 could reduce it may prove it to be a good biomarker of post-COVID depression. Unfortunately, so far only one study mentions the relationship between BDNF and COVID-19, but its results are promising and it is worth doing more research in this direction.

## 4. Discussion

In recent years, the cause of MDD has been increasingly researched, and one of the leading hypotheses is an active inflammatory process with elevated pro- and anti-inflammatory cytokines and oxidative stress [226]. Increased levels of inflammatory factors are associated with excessive activation of the kynurenine pathway and, as a result, reduced levels of tryptophan and serotonin, as well as an excess of neurotoxic metabolites such as kynurenine, quinolinic acid or 3-hydroxykynurenine. Inflammatory factors and chronic stress are also reflected in the activity of the HPA axis and cause its hyperactivity, which may contribute to the decline in BDNF [226]. All these accumulating changes cause neurotoxicity, neurodegeneration, inhibition of neurogenesis, disorders of synaptic plasticity and the structure of synaptic membranes, which all together result in the occurrence of MDD. 

In our study, we noticed that many of the postulated depression biomarkers [32,148] also appear in COVID-19 (+) patients. So far, the most described and most significant are IL-6, sIL-6R, IL-10, IL-1β, TNFα, sTNFR1, sTNFR2, IFN-γ, CRP, IL-2/sIL-2R, SAA1, BDNF, kynurenine, quinolinic acid, 3-hydroxykynurenine, and reduced tryptophan and BDNF. Considering the above-mentioned biomarkers, we hypothesize that the cascade leading to the development of post-COVID depression is analogous to the inflammatory depression hypothesis mentioned above. Infection with SARS-CoV-2 virus causes the body’s immune response in which pro-inflammatory cytokines such as TNF-α, IL-6, IL-1β, IL-2 and IFN-γ are secreted. They contribute to increasing the permeability of BBB, and this, combined with their elevated peripheral levels, causes their magnified penetration into the brain. In addition, factors such as MCP-1 cause an increased influx of immune system cells into the regions associated with MDD and additional cytokine secretion at this site. A high concentration of cytokines in the brain, including its strongest activator—IFN-γ, contributes to the activation of IDO-1 and a decrease in the synthesis of 5-HT from TRP and the accumulation of neurotoxic metabolites of the kynurenine pathway, i.e., KYN, QA, 3-HK, as well as to increased activity of the HPA axis and a decrease in BDNF production. In response to pro-inflammatory cytokines, IL-10, CRP and SAA1 are produced, the latter acting through TLR2 and TLR4 receptors present on, inter alia, astrocytes and microglia, causes further production of pro-inflammatory cytokines and worsens neuroinflammation. In addition, mechanism of viral entry into cells by using ACE2 results in its downregulation, both in the brain—where it contributes to the decline in BDNF levels, and in the gut—where it can disrupt tryptophan absorption. Accumulating cytokines, inflammatory factors, toxic metabolites, oxidative stress mediators and a decrease in BDNF, tryptophan and serotonin levels cause disturbances in neurotransmission, induce apoptosis of nerve cells and negatively affect synaptogenesis, synaptic plasticity and hippocampal neurogenesis. Demonstrated in some studies is the long-lasting increased level of such cytokines as, for example, IL-6, IL-1β and TNF-α, which means a longer exposure to factors contributing to the development of depression. High concentrations of the aforementioned cytokines may also cause changes in brain structures associated with MDD. Elevated IL-6 and TNF-α have been shown to increase the activity of the amygdala—the region associated with anxiety and depressive symptoms. The rise in amygdala activity results in increased production of inflammatory cytokines [227]. A separate study on woman in grief indicated elevated levels of IL-1β and sTNFR-2 in saliva and their correlation to the activation of subgenual anterior cingulate cortex—the region that plays a role in regulating emotions and connected to anhedonia in MDD. Similar changes in brain substrates may occur in post-COVID depression [228]. Delayed onset of MDD in COVID-19 survivors might also be related to phenomenon of neuroinflammatory priming which is defined as alteration of subsequent neuroinflammatory response to immune challenges caused by prior stress or immune activation (often triggered by infection). The ‘First hit’ which would be the SARS-CoV-2 infection may not necessarily cause psychiatric symptoms, yet it might elevate sensitivity and exaggerate the immunological response to other pro-inflammatory agents e.g., mild infections, injuries, psychological stress. The exact mechanism of this phenomenon has not yet been well understood [229]. 

There is evidence of an increased incidence of MDD in people suffering from inflammatory diseases such as multiple sclerosis or systemic lupus, and in those treated with cytokines. The prolonged state of sustained high levels of inflammatory factors and a cytokine storm in COVID-19 may therefore have the same or even greater impact on the development of MDD. Patients with a severe course of the disease are particularly vulnerable, as they had the highest and longest-lasting levels of the aforementioned cytokines and inflammatory factors.

## 5. Conclusions

The biomarkers described in this review were the most frequent appearances in other studies on COVID-19 and common with the so far proposed MDD biomarkers (Table 1).

However, there is no evidence that any individual factor can serve as a biomarker of MDD [230], so we conclude that in order to assess the risk of developing and diagnosing it, a more holistic post-COVID depression biosignature study should be performed, taking into account all the factors listed here. In a disease as severe and turbulent as COVID-19, the assessment of biomarkers for MDD can be technically difficult to perform and disrupted by disorders that occur, in case of this illness, in most body systems. However, the described high percentage of patients who develop depression, combined with the huge numbers of patients hospitalized due to COVID-19, make the search for biomarkers of this disease, and thus faster diagnosis and more effective treatment, of great importance for mental health on a global scale. Unfortunately, there is still little research on this subject and it is often inconclusive. There is also a lack of comprehensive studies directly linking MDD and its biomarkers to COVID-19. In future, in order to assess the true usefulness and relationship of the post-COVID depression biomarkers described in this study, there is a need for a prospective study linking their baseline level with the subsequent development of MDD. 

## Figures and Tables

**Figure 1 jcm-10-04142-f001:**
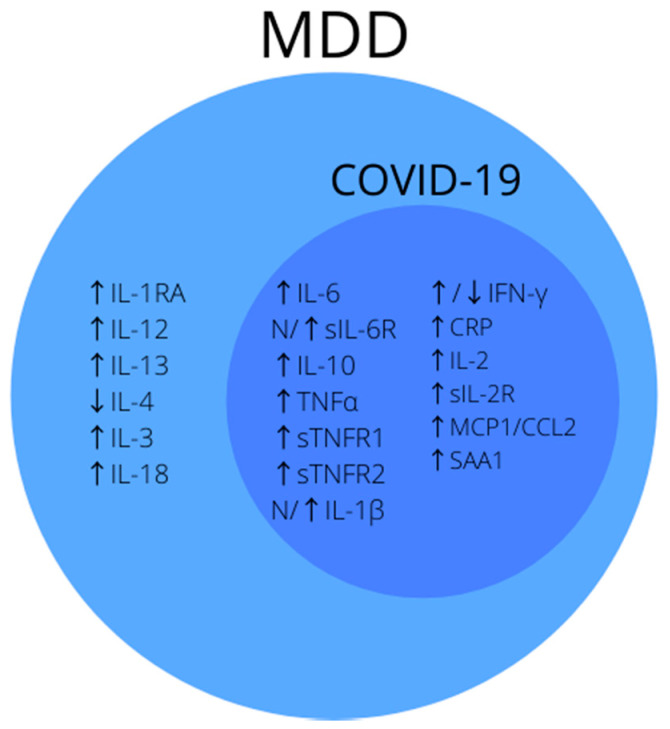
Summary of the most common inflammatory biomarkers of MDD. Note that biomarkers inside dark-blue circle coincide with markers of disturbed homeostasis in COVID-19. Abbreviations: ↑—increased concentration; ↓—decreased concentration; N—normal concentration; CRP—C-reactive protein; IL-1—interleukin-1; IL-2—interleukin-2; IL-3—interleukin-3; IL-4—interleukin-4; IL-6—interleukin-6; IL-10—interleukin-10; IL-12—interleukin-12; IL-13—interleukin-13; IL-18—interleukin-18; IL-1RA—interleukin-1 receptor antagonist; IFN-γ—interferon γ; MDD—major depressive disorder; MCP-1/CCL2—Monocyte Chemoattractant Protein-1/Chemokine ligand 2; sIL-2R—soluble interleukin 2 receptor; sIL-6R—soluble interleukin 6 receptor; sTNFR-1—soluble tumor necrosis factor α receptor 1; sTNFR-2—soluble tumor necrosis factor α receptor 2; SAA1—serum amyloid a; TNF-α—tumor necrosis factor α.

**Figure 2 jcm-10-04142-f002:**
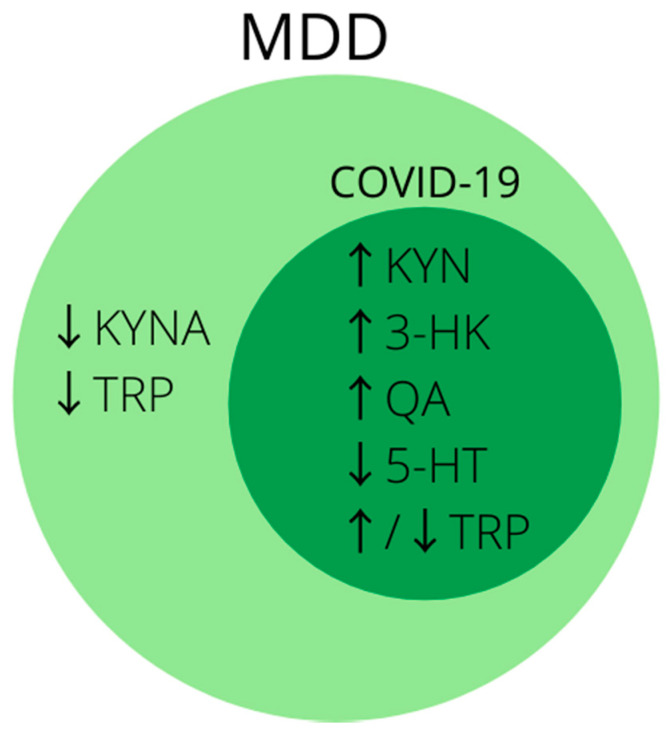
Summary of the most common biomarkers of activated kynurenine pathway in MDD. Note that biomarkers inside dark-green circle coincide with markers of disturbed tryptophan metabolism in COVID-19. Abbreviations: ↑—increased concentration; ↓—decreased concentration; 3-HK—3-hydroxykynurenine; 5-HT—serotonin; KYN—kynurenine; KYNA—kynurenic acid; MDD—major depressive disorder; QA—quinolinic acid; TRP—tryptophan.

**Figure 3 jcm-10-04142-f003:**
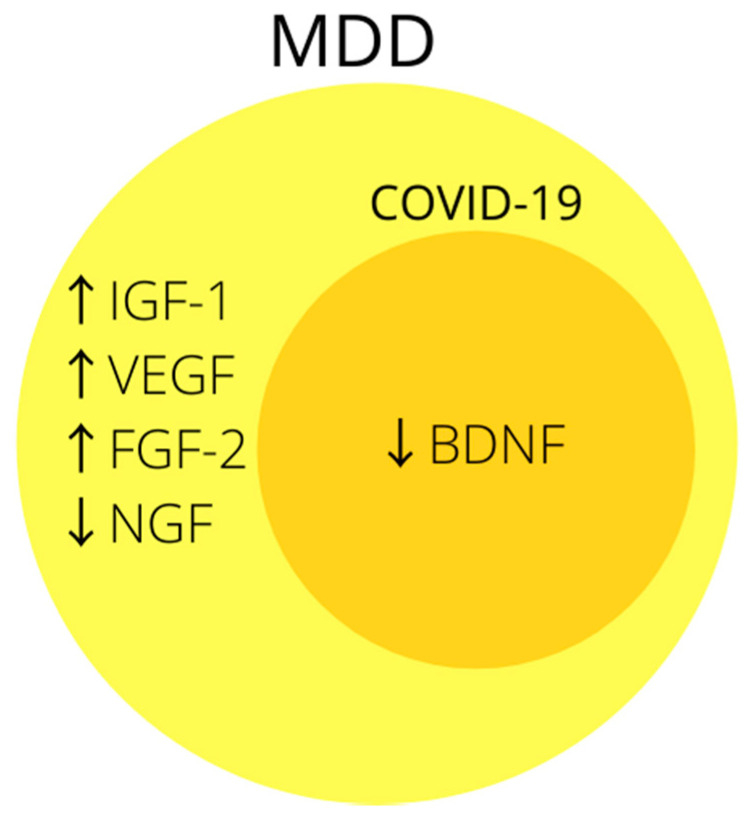
Summary of the most common changes in growth factors in MDD. Note that biomarkers inside dark-yellow circle coincide with changes in growth factors in COVID-19. Abbreviations: ↑—increased concentration; ↓—decreased concentration; BDNF—brain derived neurotrophic factor; FGF-2—fibroblast growth factor-2; IGF-1—insulin-like growth factor-1; NGF—nerve growth factor; VEGF—vascular endothelial growth factor.

**Table 1 jcm-10-04142-t001:** Common biomarkers of MDD compared to findings in COVID-19 (+) patients. Abbreviations: ↑—increased concentration; ↓—decreased concentration; N—normal concentration; -—not documented; 3-HK—3-hydroxykynurenine; 5-HT—serotonin; BDNF—brain derived neurotrophic factor; CRP—C-reactive protein; FGF-2—fibroblast growth factor-2; IGF-1—insulin-like growth factor-1; IL-1—interleukin-1; IL-2—interleukin-2; IL-3—interleukin-3; IL-4—interleukin-4; IL-6—interleukin-6; IL-10—interleukin-10; IL-12—interleukin-12; IL-13—interleukin-13; IL-18—interleukin-18; IL-1RA—interleukin-1 receptor antagonist; IFN-γ—interferon γ; KYN—kynurenine; KYNA—kynurenic acid; MDD—major depressive disorder; MCP-1/CCL2—Monocyte Chemoattractant Protein-1/Chemokine ligand 2; NGF—nerve growth factor; QA—quinolinic acid; sIL-2R—soluble interleukin 2 receptor; sIL-6R—soluble interleukin 6 receptor; sTNFR-1—soluble tumor necrosis factor α receptor 1; sTNFR-2—soluble tumor necrosis factor α receptor 2; SAA1—serum amyloid a; TNF-α—tumor necrosis factor α; TRP—tryptophan; VEGF—vascular endothelial growth factor.

Biomarker	MDD	COVID-19
IL-6	↑	↑
CRP	↑	↑
TNF-α	↑	↑
IFN-γ	↑	↑/↓
sIL-6R	↑	N/↑
IL-1β	↑	N/↑
IL-1RA	↑	-
IL-10	↑	↑
IL-12	↑	-
IL-13	↑	-
IL-4	↓	-
IL-3	↑	-
IL-2	↑	↑
sIL-2R	↑	↑
sTNFR-4	↑	↑
sTNFR-2	↑	↑
IL-18	↑	-
MCP-1/CCL2	↑	↑
SAA1	↑	↑
KYN	↑	↑
KYNA	↓	↑
QA	↑	↑
3-HK	↑	↑
TRP	↓	↑/↓
5-HT	↓	↓
BDNF	↓	↓
IGF-1	↑	-
NGF	↓	-
VEGF	↑	-
FGF-2	↑	-

## Data Availability

Data sharing not applicable. No new data were created or analyzed in this study. Data sharing is not applicable to this article.

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
