# Peer review of "Biomarkers of Post-COVID Depression"

_jcm, 2021, doi:10.3390/jcm10184142_

Round 1
Reviewer 1 Report
- What is MDD? Need to explain it.
- Need more explaintion what you talk about the purpose of study and need for this resesarch (e.g., potential correlation between biomarkers of depression and markers increased in COVID-19) in introduction section.
- A detailed explanation of the research method (e.g., research procedure, classification criteria et al.) is required.
- Provide characteristics of each section in results (by tables or graph).
- What are conclusions and recommendations of this study?
- The overall English needs to be corrected.
Author Response
Dear Reviewer,
Thank you for your comments.
- What is MDD? Need to explain it.
Line 56 - corrected
Depression, also called major depressive disorder (MDD)
- Need more explaintion what you talk about the purpose of study and need for this resesarch (e.g., potential correlation between biomarkers of depression and markers increased in COVID-19) in introduction section.
Line 73-84 - elaborated
The aim of the study is to find potential link between biomarkers of MDD and markers of disturbed homeostasis of the organism during or after COVID-19. Common links of those two states can be of use as prognostic biomarkers of post-COVID depression and indicate the cause of MDD development in those patients. The acquired knowledge could be used in the future to determine which COVID-19(+) patients are at risk of developing MDD or making a more confident diagnosis in those who are already affected. This will allow to identify the group of patients in whom mental health should be paid special attention in the phase of recovery from illness and during regaining social functionality from before SARS-CoV-2 infection. Doing so would enable to implement effective treatment faster which will reduce recovery time for patients, reduce treatment costs and economic burden of COVID-19 and contribute to some extent to reducing the global mental health problem
- A detailed explanation of the research method (e.g., research procedure, classification criteria et al.) is required.
Line 86-98 - elaborated
A literature search was conducted in PubMed, Scopus and Google Scholar databases. We included clinical studies, reviews, meta-analyzes and case studies regarding depressive symptoms in COVID-19 (+) patients. The search strategy consisted of the following keywords: ‘depression’, ‘biomarker’, ‘COVID-19’, ‘SARS-CoV-2’, ‘post-COVID’, ‘long-COVID’, ‘metabolomic’, ‘inflammatory’, ‘immunological’, ‘endocrinological ‘ ‘oxidative stress’, ‘HPA axis’, ‘neurotrophic’, ‘biosignature’ as well as combinations of these terms. We then excluded all articles in which MDD was described in the context of the social consequences of a pandemic such as isolation, stress, fear of disease or economic problems, in order to be able to analyze only the direct impact of SARS-CoV-2 on the body and the development of post-COVID depression. Relevant studies were then included with the intention of covering the widest possible spectrum of different markers for post-COVID depression. We conducted additional manual searches of the references of the related articles in order to gather information about the relevant supporting literature
- Provide characteristics of each section in results (by tables or graph).
Line 136, 419 and 448. We added figures summarizing sections.
- What are conclusions and recommendations of this study?
Line 516. We singled out a section of conclusions with the recommendations for future studies.
- The overall English needs to be corrected.The manuscript was read by English language native speaker and we changed it according to his comments.
Reviewer 2 Report
This is a very well-written and timely review that covers all the major types of inflammatory markers modulated during COVID19. I have some minor comments:
Section 3.1.1- Please begin the section by introducing IL-6/sIL-6R. In its current form, it reads a bit odd. Same with 3.1.4.
However, reports on the role of IFN-γ are not unequivocal. Some studies showed an increase in the level of gamma interferon among people with depression- isn't IFN-γ the same as gamma interferon? Please clarify and follow a consistent nomenclature.
Please change papers to studies.
Please expand IL-2 when introduced for the first time.
The authors alternate between MDD and depression. This is a bit confusing. Please stick to a single nomenclature for clarity to readers.
In the very first section regarding COVID it will be nice to mention some statistics on global healthcare expenses to reiterate the enormity of the crisis.
Line 393: we hipotetize that the cascade- Please correct the spelling for hypothesize.
The authors must reflect a bit more in their conclusive remarks regarding how such inflammatory storms during and after COVID19 modulates brain substrates leading to MDD. How do these persistent inflammatory drives alter the neuroadaptations chronically to influence MDD later in life etc
Line 189: IL1 beta is provided as IL-1B. Please be consistent.
Author Response
Dear Reviewer,
Thank you for your comments.
Section 3.1.1- Please begin the section by introducing IL-6/sIL-6R. In its current form, it reads a bit odd. Same with 3.1.4.
We expanded on IL-6, IL-1B according to your suggestions and added a bit more on IL-10.
Line 149:
Interleukin 6 (IL-6) is a pro-inflammatory pleiotropic cytokine secreted mainly by monocytes and macrophages under the influence of interleukin 1β and TNF-α, but also by astrocytes and microglia. It belongs to the family of proteins that use gp130 as a signal transmitter [64]. It works by inducing the differentiation of activated B cells towards antibody-secreting cells, stimulating the synthesis of acute phase proteins such as C-reactive protein, serum amyloid A, fibrinogen, α1-antitrypsin and haptoglobin in the liver, and promoting the differentiation of naive CD4 + T cells [65]. It also plays a role in the body as a mediator of a warning signal about tissue damage or other sudden events. Its level increases in the event of infection, inflammation or trauma [64]. IL-6 also affects the functionality of neurons, may disrupt hippocampal neurogenesis and intensifies neuroinflammation [66].
Line 167:
According to some studies, high levels of IL-6 correlate with the severity of MDD symptoms in patients who do not respond to treatment [70]. It has been observed that in patients with MDD, increased levels of IL-6 correlated with attention deficit disorder, and one study showed that its increase was prior to the incidence of cognitive impairment in patients with MDD [69].
Line 180:
Interleukin 10 (IL-10) is an anti-inflammatory cytokine produced by Th2, Treg cells and M2 macrophages [76]. In the central nervous system (CNS) it is produced, inter alia, by astroglia and microglia [77]. In the latter, the secretion of IL-10 is augmented by neurotransmitters and damage-associated molecules – glutamate and adenosine respectively [78]. Overall IL-10 production increases with increased levels of IL-6 and TNF-α [79,80]. It exerts its function by binding with IL-10 receptor (IL-10R) which consist of two subunits – IL-10R1 and IL-10R2. The latter is expressed in most cell types, but IL-10R1 is mostly restricted to cells of hematopoetic lineage. Due to the myeloid origins, microglia express both subunits of IL-10R. Unexpectedly, resting astrocytes also express IL-10R1 [78]. IL-10 limits neuroinflammation, promotes production of immunosuppressive transforming growth factor β (TGF-β) by astrocytes and reduces astrogliosis in response to the pathogenic factors [78].
Elevated levels of IL-10 are often reported in people with MDD [81–83]. Few researchers also observed that MDD severity is related to increased IL-10 [84,85]. Two meta-analyzes have shown that IL-10 level decreases with effective antidepressant treatment [81,86].
Line 234:
Interleukin 1β (IL-1β) is another pro-inflammatory cytokine secreted by the same cell types as TNF-α and IL-6, i.e. Th1 lymphocytes and M1 macrophages. In the CNS it can be secreted by different types of cells, e.g. astroglia, oligodendrocytes, neurons and microglia, furthermore, they all express IL-1β receptors [33,116]. Its effects on the brain are, as in the case of TNF-α and IL-6, induction of apoptosis [117,118] and negative effects on synaptogenesis and synaptic plasticity [119,120]. Few studies proved that physiological levels of IL-1β promotes long term potentiation (LTP) and acquisition and retention of memory. In the other hand, its pathophysiological levels as seen in inflammation can disturb LTP and cause failure of memory acquisition and its recall [116].
However, reports on the role of IFN-γ are not unequivocal. Some studies showed an increase in the level of gamma interferon among people with depression- isn't IFN-γ the same as gamma interferon? Please clarify and follow a consistent nomenclature.
-corrected
However, reports on the role of IFN-γ are inconsistent. Some studies demonstrated its increase among people with MDD [143,144,146], and some showed no increase or even a decrease [46,137].
Please change papers to studies.
-corrected
Please expand IL-2 when introduced for the first time.
- corrected
The authors alternate between MDD and depression. This is a bit confusing. Please stick to a single nomenclature for clarity to readers.
- we corrected most 'depression' to MDD but in order to avoid many repetitions in some fragments of the text we had to leave few. For the clarity of readers we added an explanation in text. Line 56: "Depression, also called major depressive disorder (MDD)"
In the very first section regarding COVID it will be nice to mention some statistics on global healthcare expenses to reiterate the enormity of the crisis.
Line 49-55. There is lack on information on global healthcare expenses but we added a fragment on economic burden in US alone
"The COVID-19 crisis is about more than general health problems. Prolonged disease symptoms, employees' inability to work, shutting down businesses, premature deaths, cost of COVID-19 treatment and prevention make most of the countries affected by the pandemic feel the economic crisis as well. In the United States alone the total cost of the pandemic is estimated at approximately 90% of the annual gross domestic product, which is more than $16 trillion [14]. Therefore, the search for different ways to limit the effects of pandemic is of great importance for the economic sector as well."
Line 393: we hipotetize that the cascade- Please correct the spelling for hypothesize.
- corrected
The authors must reflect a bit more in their conclusive remarks regarding how such inflammatory storms during and after COVID19 modulates brain substrates leading to MDD. How do these persistent inflammatory drives alter the neuroadaptations chronically to influence MDD later in life etc
- expanded
"High concentrations of the aforementioned cytokines may also cause changes in brain structures associated with MDD. Elevated IL-6 and TNF-α have been shown to increase the activity of amygdala – region associated with anxiety and depressive symptoms. The rise in amygdala activity results in increased production of inflammatory cytokines [230]. A separate study on woman in grief indicated elevated levels of IL-1β and sTNFR-2 in saliva and their correlation to the activation of subgenual anterior cingulate cortex – region that plays a role in regulating emotions and connected to anhedonia in MDD. Similar changes in brain substrates may occur in post-COVID depression [231]. Delayed onset of MDD in COVID-19 survivors might also be related to phenomenon of neuroinflammatory priming which is defined as alteration of subsequent neuroinflammatory response to immune challenges caused by prior stress or immune activation (often triggered by infection). ‘First hit’ which would be SARS-CoV-2 infection may not necessarily cause psychiatric symptoms yet it might elevate sensitivity and exaggerate the immunological response to other pro-inflammatory agents e.g. mild infections, injuries, psychological stress. The exact mechanism of this phenomenon has not yet been well understood [232]."
Line 189: IL1 beta is provided as IL-1B. Please be consistent.
- corrected
Round 2
Reviewer 1 Report
Thank you for your revision, so I accept in this paper. Have a good one.